# Comparison of the Hemodynamic Performance of Two Neuromuscular Electrical Stimulation Devices Applied to the Lower Limb

**DOI:** 10.3390/jpm10020036

**Published:** 2020-05-07

**Authors:** Sahar Avazzadeh, Andrea O’Farrell, Kate Flaherty, Sandra O’Connell, Gearóid ÓLaighin, Leo R. Quinlan

**Affiliations:** 1Physiology, School of Medicine, NUI Galway, University Road, Galway H91 TK33, Ireland; sahar.avazzadeh@nuigalway.ie (S.A.); A.OFARRELL2@nuigalway.ie (A.O’F.); K.FAHERTY4@nuigalway.ie (K.F.); s.oconnell10@nuigalway.ie (S.O’C.); leo.quinlan@nuigalway.ie (L.R.Q.); 2Electrical & Electronic Engineering, School of Engineering, NUI Galway, University Road, Galway H91 TK33, Ireland; 3Human Movement Laboratory, CÚRAM—Centre for Research in Medical Devices, NUI Galway, University Road, Galway H91 TK33, Ireland

**Keywords:** venous leg ulcer, neuromuscular electrical stimulation, lower limb hemodynamic

## Abstract

Currently, 1% of the population of the Western world suffers from venous leg ulcers as a result of chronic venous insufficiency. Current treatment involves the use of moist wound healing, compression bandages, and intermittent pneumatic compression. Neuromuscular electrical stimulation is a novel potential new therapeutic method for the promotion of increased lower limb hemodynamics. The aim of this study was to measure the hemodynamic changes in the lower limb with the use of two neuromuscular electrical stimulation devices. Twelve healthy volunteers received two neuromuscular stimulation device interventions. The Geko^TM^ and National University of Ireland (NUI) Galway neuromuscular electrical stimulation devices were randomized between dominant and non-dominant legs. Hemodynamic measurements of peak venous velocity (cm/s), the time average mean velocity (TAMEAN) (cm/s), and ejected volume (mL) of blood were recorded. Peak venous velocity was significantly increased by the Geko^TM^ and the NUI Galway device compared to baseline blood flow (*p* < 0.0001), while only the voluntary contraction produced significant increases in TAMEAN and ejected volume (both *p* < 0.05). Neuromuscular muscular electrical stimulation can produce adequate increases in lower limb hemodynamics sufficient to prevent venous stasis. Greater use of neuromuscular stimulation devices could be considered in the treatment of conditions related to chronic venous insufficiency but requires further research.

## 1. Introduction

There is an increasing global incidence of people presenting with venous leg ulcers (VLUs) [1]. VLUs occur as a result of sustained venous hypertension in the lower leg, which is brought on by chronic venous insufficiency (CVI). CVI develops due to a number of related conditions that affect the hemodynamics in the lower leg, but there is no distinct pattern in which they occur [2]. One of these factors includes an increased hydrostatic pressure as a result of venous reflux in the superficial veins, due to valve dysfunction. There can also be damage resulting from thrombosis, weakness in the vascular walls, or deformities, which affect deep veins, leading to valve incompetence. Another factor that can contributes is chronic muscle or joint inflammation resulting in a reduced range of motion, leading to underactivation of the calf muscle pump, and resulting in poor venous outflow, thus leading to prolonged venous stasis [3,4]. Patients presenting with persistent VLUs have a decreased quality of life as a result of chronic pain, in addition to a risk of complications, such as infection, cellulitis, and other conditions [4]. Recent figures predict that 1% of the population in the Western world suffer from VLUs, with the full economic cost of VLUs estimated at $15 billion annually to the health budget of the USA alone [5,6].

The 2009, the Effect of Surgery and Compression on Healing and Recurrence (ESCHAR) trial showed that surgical ablation of superficial veins reduced ulcer recurrence, but surgery did not improve healing [7]. The most recent guidelines from the European Society for Vascular Surgery (ESVS) in 2015 recommend compression as a key element of venous disease management. The more recent Early Venous Reflux Ablation (EVRA) trial provides clear evidence that ablation surgery accelerates healing and reduces recurrence; however, this approach may not be applicable in all cases [8] and longer-term follow-up data is required before definitive guidance can be revised. Thus, currently, compression is still a cornerstone of venous therapy and this paper explores a potential adjunct that may enhance ulcer healing. Thus, the mainstay treatment for VLUs remains the use of moisture retentive dressings to reduce healing time, graduated compression bandages, and in some cases, intermittent pneumatic compression (IPC) to offset the venous hypertension by squeezing the lower leg to force pooled blood out of the lower limbs [9,10]. Patients are advised to take physical exercise, such as walking, to activate the calf muscle pump to help increase lower leg hemodynamics and prevent venous stasis [11,12,13]. 

The main goal of treatment is to counteract elevated venous pressure in the lower leg, increase venous return from the lower limbs, and return normal hemodynamics. The best method for increasing venous return is through activation of the calf muscle pump. Contraction of the calf muscle causes the veins in the lower leg to be squeezed, causing blood in these veins to be expelled, and preventing venous stasis [14]. Contraction also leads to increased arterial and microcirculatory flow and the reduction of pressure within the limb, preventing edema [3]. A common challenge for patients with VLUs is the ability to mobilize or to adhere to treatment plans. Additionally, trained medical professionals are required for the application of dressings and compression bandages, which can be inconvenient for a patient. This results in missing appointments and leading to the VLU being in a state of non-healing or further deterioration. With the ‘gold standard’ of wound care, only 70% of VLUs recover within 3 months, so 30% fail to heal within this time frame and are known as a chronic wound [15,16].

More recently, there has been a focus on novel treatment options for VLUs with the use of neuromuscular electrical stimulation (NMES) devices as an adjunct therapy to the standard care [5,6,16]. This is achieved by placing skin surface electrodes over the motor points on the lower leg, leading to an electrical stimulus passing through these electrodes. This generates either an action potential in the nerve causing contraction of the calf muscle or the stimulus can directly excite muscle units without nerve activation [17]. This is an effective method in the treatment of VLUs since activation of the calf muscle pump is the most effective way of ejecting pooled blood from the lower leg [18,19,20,21]. This activation by an NMES device leads to an ejection volume from the venous compartment that is only slightly less than a voluntary contraction [22]. The main attraction of NMES devices is that they are commonly portable and easily operated by a patient or their carer, and this could lead to reduced dependency on medical professionals as it can be used in a home environment. In comparison to intermittent pneumatic compression (IPC), the majority of NMES devices are quiet and well tolerated by users.

In this study, we explored the capability of two portable NMES devices to improve the hemodynamic performance of the lower limb in young healthy adults.

## 2. Materials and Methods

Subjects: Twelve young healthy subjects (7 male and 5 female) were recruited for this study. The study group had a mean age (years) of 22.17 ± 1.99, body weight (kg) 70.93 ± 13.92, height (m) 1.72 ± 0.11, BMI (kg/m^2^) 23.78 ± 2.59, and with no history of cardiovascular disease, high blood pressure, or deep vein thrombosis (DVT). Ethical approval was obtained from the Research Ethics Committee (09/SEP/04), NUI Galway, and all subjects provided written consent and were fully aware of all procedures. 

Study protocol: Subjects were seated on a physiotherapy table in a semi-recumbent position. Both the dominant and non-dominant leg of each subject were tested for hemodynamic performance. Four test conditions of a 4-min duration were applied to each subject: Control (baseline at rest), voluntary contraction, soleus muscle NMES, and peroneal nerve Geko^TM^ stimulation. Three interventions were applied to a leg of each subject: Baseline (rest), voluntary muscle contraction, and one of the NMES devices. Device placement on each leg was randomized before the study began, with each device having an equal chance of being placed on the either leg. After each intervention, a rest period of 3 min was allowed to ensure baseline equilibrium venous flow was achieved. Volume flow was measured after each rest period for 1 min to ensure baseline blood flow was returned to normal. 

Voluntary contraction protocol: All subjects were required to perform a voluntary contraction of the calf muscle. Subjects were in a standing position and instructed to move to a maximal ‘tiptoe’ position. This produces an exaggerated calf muscle contraction. The subject was then asked to hold for 2 s at the maximum contraction and then return to the starting position. When the subject’s heels touched the floor, a 20-s rest period was given. This was repeated over a 4-min period, with 4 voluntary contractions being captured. This was carried out in such a manner to match the NMES timing protocol. 

Geko^TM^ device protocol: This T-1 model (Geko^TM^, FirstKind Ltd., High Wycombe, UK) is supplied as a self-adhesive disposable 24-h lifetime model that applies a 27 mA constant current output and has 7 different pulse width settings to choose from, ranging from 70–560 μs, with a pulse frequency of 1 Hz. The Geko^TM^ device was attached to the skin over the fibular head as per the manufacturer’s guidelines. This placement allows for stimulation of the common peroneal nerve before branching into the deep and superficial peroneal nerves. Stimulation of these nerves activates the peroneus longus, peroneus brevis, tibialis anterior, extensor hallucis longus, extensor digitorum longus, peroneus tertius, and extensor digitorum brevis muscle groups. The T-1 was set at a level that led to a full activation of the above muscle groups, seen as an obvious foot twitch. The mean stimulation level used on each participant was 4. 

NUI Galway device protocol: The NUI Galway NMES device (Electrical & Electronic Engineering, NUI Galway, Galway, Ireland) uses two self-adhesive 5cm × 5 cm PALS surface electrodes (Ultrastim, Axelgaard Manufacturing Co. Ltd. Fallbrook, CA, USA) that were placed over the motor points of the soleus muscles on the designated leg as described previously [20]. To ensure correct electrode placement and to ensure that the subject was comfortable with the sensation of electrical stimulation, a series of test pulses were applied initially at a very low intensity for sensory stimulation. The stimulation intensity was gradually increased until the motor threshold was reached, and a first visible contraction of the soleus muscle was observed. This was indicated as a visible tightening of the soleus muscle and plantarflexion of the foot. After reaching the motor threshold, the intensity was further increased until the maximum tolerable stimulation intensity was achieved. This intensity was used for the remainder of the NMES protocol. The mean stimulation intensity voltage used on all the participants was 17 V. The NUI Galway device was set to a total cycle time of 23 s, with an OFF time of 20 s, ramp-up time of 1 s, a contraction time of 1 s, a ramp-down time of 1 s, a pulse frequency of 36 Hz, and a balanced biphasic waveform with a pulse width of 350 μs. These stimulation parameters were selected to provide an effective contraction, while maximizing subject comfort based on previous work using NMES [20,23].

Hemodynamic measurement: Lower leg hemodynamic measurements were recorded with a Duplex Doppler ultrasound using a 4–8 MHz linear transducer (LOGIQ e; GE Medical Systems, Dublin, Ireland). The blood flow measurements were taken from the popliteal vein at the popliteal fossa at the back of the knee to reflect the venous outflow from the deep veins in the lower leg. Blood flow for each intervention was measured 4 times, with no measurement being taken for the first minute of each intervention. The samples were then taken during the remaining intervention period at minute intervals on 2, 3, and 4, while the voluntary contraction was measured during the contraction of the calf muscle. The size of each measurement window for all interventions was set at 12 s. The measurement window for all test conditions, included 1 s before stimulation, and 11 s of blood flow post stimulation onset (Figure 1). The diameter (cm) of the popliteal vein was measured from the Doppler image. Using the Doppler waveform, the peak venous velocity (PV) (cm/s) measurement was recorded. PV is a measurement parameter used to measure the effectiveness of the DVT prevention methods. Increases in PV are seen to be beneficial for the prevention of DVT [24,25,26]. The blood flow response to the intervention (NMES, voluntary contraction) is clearly distinguishable from the waveform. The time-averaged mean velocity of blood flow (TAMEAN) (cm/s) and volume flow (mL/min) were calculated by the Doppler units installed software. Volume flow was calculated as the product of the TAMEAN (cm/s) **×** the measured cross-sectional area of the popliteal vein (CSA) (cm^2^). Increases in TAMEAN indicate an increased blood flow velocity through a vein being viewed using an ultrasound device. This would suggest that there is reduced venous pooling as a result of an increased TAMEAN. 

Ejected volume (EV) is the measurement of the volume of blood displaced through the popliteal vein during a single pulse of the intervention. Increases in EV indicate less venous pooling as the blood is displaced from the compartment it had pooled in and this would be beneficial in the treatment of venous stasis. This was calculated using the following equation:

Ejected Volume (mL) = Volume flow (mL/min) × Duration of single intervention pulse(min).

The duration of a single intervention pulse represents the time for one pulse of the intervention (NMES stimulation) defined by the NMES protocol above. For the baseline and voluntary contraction, the duration of intervention pulse was taken as 1 s. Representative waveforms for each condition are shown in Figure 1. 

Statistical analysis: Statistical analysis was carried out using a one-way ANOVA (Tukey multiple comparison test). This was performed using GraphPad Prism. The significance was taken at *p* < 0.05. 

## 3. Results

### 3.1. Peak Velocity Significantly Enhanced by Voluntary Contractions and NMES 

The PV for baseline, voluntary contraction, NUI Galway, and Geko^TM^ devices were 17.54 cm/s ± 4.1, 75.91 cm/s ± 13.8, 63.82 cm/s ± 18.2, and 46.34 cm/s ± 18.3, respectively. All interventions were significantly greater than baseline (*P* < 0.0001). Additionally, the PV for voluntary contractions was significantly greater than that produced by the Geko^TM^ devise (*P* < 0.0001), which in turn was significantly greater than that recorded with the NUI Galway device (*P* = 0.0175) (Table 1, Figure 2). 

### 3.2. TAMEAN Significantly Enhanced by Voluntary CONTRACTIONS only. 

For the time average mean velocity (TAMEAN), only the voluntary contraction resulted in a significant change from baseline (*P* < 0.0001). The TAMEAN was 3.57 cm/s ± 1.0, 7.17 cm/s ± 3.0, 5.34 cm/s ± 1.6, and 4.20 ± 1.6 for the baseline, voluntary contraction, NUI Galway, and Geko^TM^ devices, respectively (Figure 3). While the voluntary contraction resulted in the most significant increase in TAMEAN, neither the Geko^TM^ (*P* < 0.86) nor the NUI Galway (*P* < 0.08) interventions showed a significant increase in comparison to baseline (Figure 3). The Geko^TM^ was significantly lower compared to the voluntary contraction (*P* = 0.0025) (Table 1).

### 3.3. Ejected Volume Significantly Enhanced by Voluntary Contractions and NMES 

The voluntary contraction resulted in the most significant increase in EV compared to the baseline (*P* < 0.0004). While both the NUI Galway and Geko^TM^ devices did have modest increases in EV, neither was significant from baseline. EV was 16.62 mL ± 9.1, 30.65 mL ± 13.7, 23.14 mL ± 11.4, and 19.76 mL ± 10.6 for baseline, voluntary contraction, NUI Galway, and Geko^TM^ devices, respectively (Table 1, Figure 4). 

### 3.4. Effect of Calf Circumference on Hemodynamic Performance 

Clearly the force of muscle contraction can impact the blood flow dynamics and this force could be influenced by the calf circumference. An analysis of the calf circumference for each participant showed the mean circumference to be 35.5 cm. Subjects were stratified into two groups based on calf circumference >35 cm and calf circumference <35 cm. 

PV for calf muscle circumference < 35 cm was measured at 17.16 cm/s ± 3.8, 70.28 cm/s ± 12.7, 54.55 cm/s ± 23.7, and 61.72 cm/s ± 22.9 for baseline, voluntary contraction, NUI Galway, and Geko^TM^ devices, respectively (Figure 5). All interventions increased PV above baseline (*P* < 0.01). There was no difference between interventions, and all were equally effective in increasing PV (Table 2).

In the greater than the 35 cm group, PV values of 18.21 cm/s ± 5.1, 81.60 cm/s ± 13.5, 66.45 cm/s ± 19.2, and 40.04 cm/s ± 14.6 for baseline, voluntary contraction, NUI Galway, and Geko^TM^ devices, respectively, were found (Figure 5). Both voluntary contraction and NUI Galway increased PV above baseline (*P* < 0.001). There was no difference between voluntary contraction and the NUI Galway device. The Geko^TM^ device was not significantly different from baseline (*P* > 0.07) (Table 2).

TAMEAN for calf muscle circumference < 35 cm was recorded at 3.99 cm/s ± 0.9, 5.98 cm/s ± 1.2, 5.30 cm/s ± 1.1, and 4.66 cm/s ± 2.2 and for calf muscle circumference > 35 cm at 3.42 cm/s ± 1.1, 6.05 cm/s ± 1.4, 4.57 cm/s ± 1.4, and 4.06 cm/s ± 0.8 for baseline, voluntary contraction, NUI Galway, and Geko^TM^ devices, respectively (Figure 6). For both groups, only the voluntary contraction was significantly different from baseline (Table 1).

EV for calf muscle circumference < 35 cm was recorded at 18.05 mL ± 7.7, 27.63 mL ± 7.5, 21.46 mL ± 6.7, and 17.55 mL ± 3.7 and for calf muscle circumference > 35 cm at 18.80 mL ± 9.1, 33.93 mL ± 10.6, 24.41 mL ± 9.4, and 28.74 mL ± 11.7 for baseline, voluntary contraction, NUI Galway, and Geko^TM^ devices, respectively (Figure 7). For both groups, only the voluntary contraction was significantly different from baseline (Table 1).

## 4. Discussion

The most effective treatment to date for VLUs is the use of graduated compression bandaging, and increasingly, evidence is mounting for surgical ablation, but these methods are not without their downsides. Compression bandages can be painful for some patients, leaving them to rely on wound care and exercise alone [13]. Thus, for immobile patients, this can prove a significant negative for their recovery [27]. IPC has been proven to be effective at increasing lower limb hemodynamics and is the main alternative if a patient is unable to tolerate compression bandages [28]. There are also issues with the use of IPC, for example, the patient is unable to mobilize while using the device, the device can be loud, periods for use can be time consuming, and it has a low compliance rate. This leaves an open area for the development of new devices and methods for the treatment of VLUs. As VLUs are a result of CVI, the best treatment for VLUs is activation of the calf muscle pump through exercise as it promotes increases in hemodynamics. NMES devices offer a potential alternative treatment of CVI as NMES devices activate the calf muscle pump. 

Both the Geko^TM^ and NUI Galway devices produced an increase in PV compared to the baseline blood flow velocity, while not being as effective as a voluntary contraction. While considered less critical for VLUs, PV is a measurement commonly cited as a monitor of the effectiveness of deep vein thrombosis (DVT) prophylaxis. In patients with a high risk of DVT, PV is reduced at rest [29]. Any contraction of the calf muscle leads to an increase in PV. Both NMES devices increase PV, but to a lesser extent than a voluntary contraction. This may be beneficial as there is therefore no risk of the elicited peak venous velocities being capable of damaging the vascular walls, which would lead to an increase in venous stasis as this would result in further severity of CVI [20]. 

The TAMEAN was significantly increased by voluntary contraction only while the NUI Galway device showed a modest non-significant increase. This is an important measurement as volume flow is a product of TAMEAN and the vein cross-sectional area. Increases in TAMEAN cause an increase in volume flow. A greater volume flow would lead to a greater amount of blood moving through the popliteal vein from the lower limb, suggesting a decrease in venous stasis. The increase in TAMEAN could be a significant hemodynamic parameter to verify the effectiveness in the treatment of venous stasis with NMES devices. 

As with TAMEAN, EV was significantly increased by voluntary contraction only. Previous studies showed NMES devices to increase EV [18,23,30]. It should be noted that these studies only examined the pulse phase of the response of blood flow to the NMES device, i.e., the initial response generally in the first second of stimulation where there is a big increase in flow (Figure 1D). In this study, we used a 12-s duration of blood flow starting at the time of NMES delivery. This is a better representation of the potential effect of NMES as a therapeutic intervention of a longer time course of disease as is common with VLUs.

Typically, an increased in EV leads to less blood being left in the deep, perforator, or superficial veins, which is the main function of the calf muscle pump [31]. The Galway device produces a blood flow waveform similar to a voluntary contraction, producing a rapid pulse of blood flow followed by a quiet period and return to baseline blood flow (Figure 1B,D). The Geko^TM^ results in smaller pulses in flow per pulse, but over a 12-s measurement window, these were not significantly different from the NUI Galway device.

In segmenting the data based on the calf muscle circumference greater or less than 35 cm, we see that in the < 35 cm group, there is no difference between interventions for PV, as all were significantly greater than baseline. However, for the >35 cm group, the interventions showed differing effects with voluntary contraction and the NUI Galway device being the most effective. The profile was similar for TAMEAN and EV, but only the voluntary contraction was significant from baseline. 

## 5. Limitations

Both devices tested in this study require further examination to determine their effectiveness at altering hemodynamics to beneficial levels in terms of VLU therapy. This study is limited by the number of participants and the fact that the study was carried out on young health adults. VLUs are more commonly seen in elderly and non-ambulatory patients, as the reduced mobility seen in these groups contributes to underactivation of the calf muscle pump, which can lead to CVI [32]. A larger study with age-appropriate controls is required with these devices to determine their effectiveness on limb hemodynamics.

## 6. Conclusions

This study demonstrated that NMES devices can produce significant increases in lower limb hemodynamics. More evidence is required before NMES devices can be considered as an adjunct in the treatment of VLUs. 

## Figures and Tables

**Figure 1 jpm-10-00036-f001:**
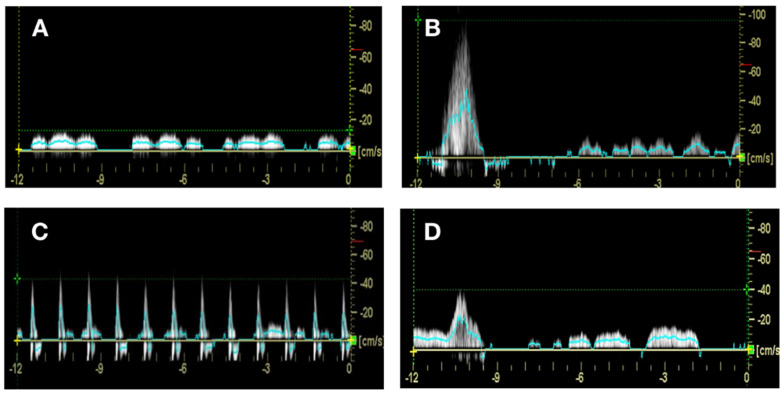
Blood flow waveforms from all interventions. Representative waveform from (**A**) baseline, (**B**) voluntary contraction, (**C**) Geko^TM^, and (**D**) NUI Galway device. The measurement window for all test conditions included 1 s before stimulation and 11 s of blood flow post stimulation onset. Peak velocity was measured off the crossing point of the blood flow waveform and the horizontal dashed line in green. The TAMEAN is plotted as a solid blue line over the course of the measurement.

**Figure 2 jpm-10-00036-f002:**
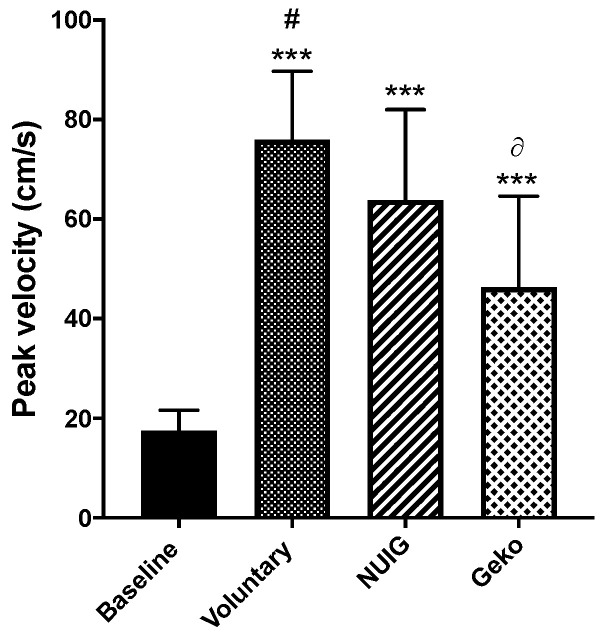
The effect of the intervention on the peak venous velocity (cm/s). The peak velocity showed a significant increase in all voluntary (*n* = 24), Geko (*n* = 9), and NUIG device (*n* = 12) interventions in comparison to baseline (*n* = 24) (****p* < 0.0001 versus. baseline; #*p* < 0.0001 versus. Geko; ∂*p* < 0.05 versus. NUIG). Data shown as ± SD.

**Figure 3 jpm-10-00036-f003:**
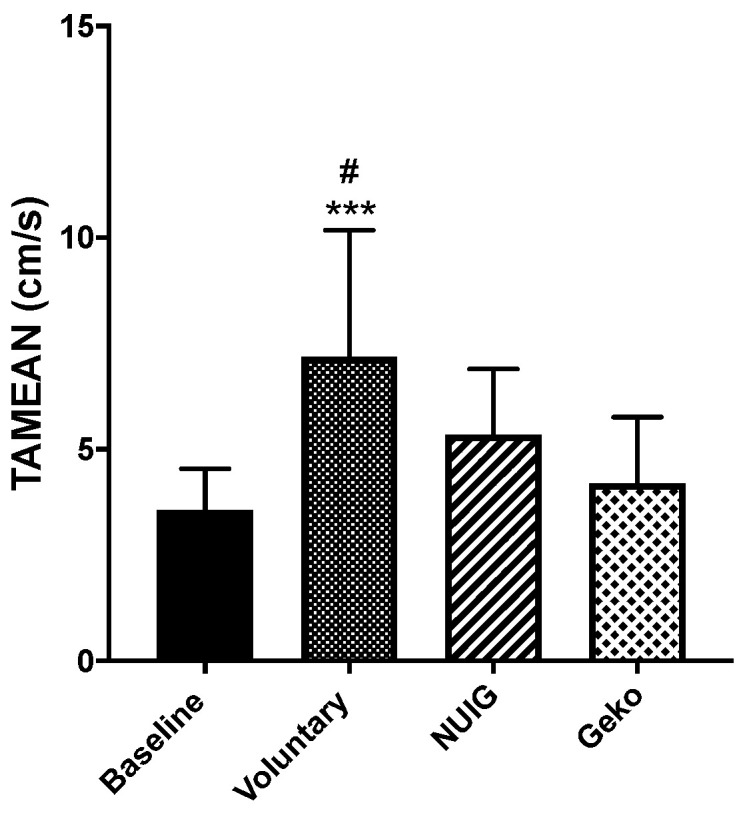
The effect of intervention on TAMEAN (cm/s). TAMEAN were significantly increased in voluntary (*n* = 24) in comparison to baseline (*n* = 24). Voluntary contraction showed significant difference with Geko (*n* = 9). (****P* < 0.0001 versus baseline; #*P* < 0.001 versus Geko) Data shown as ± SD.

**Figure 4 jpm-10-00036-f004:**
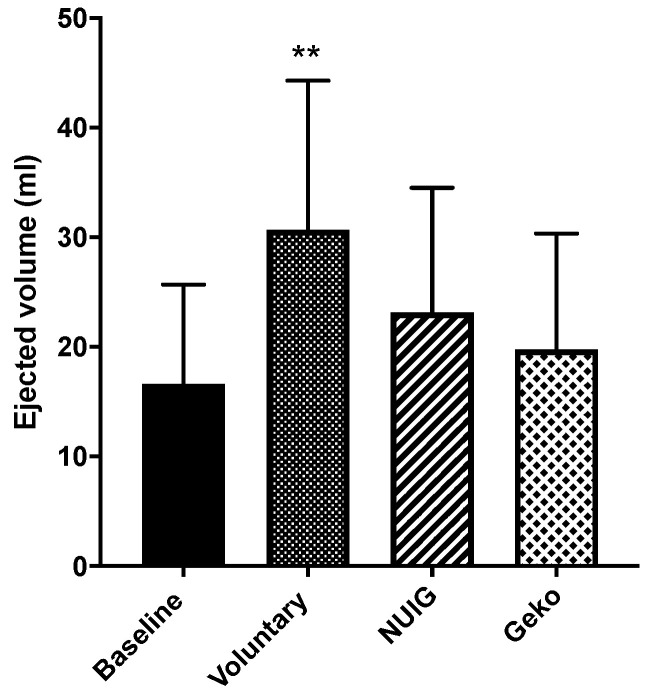
The effect of the intervention on the ejected volume (mL). The ejected volume was only significantly increased in voluntary (*n* = 24). The NUIG (*n* = 12) and Geko device (*n* = 9) showed no significant change (***P* < 0.001 versus baseline). Data shown as ± SD.

**Figure 5 jpm-10-00036-f005:**
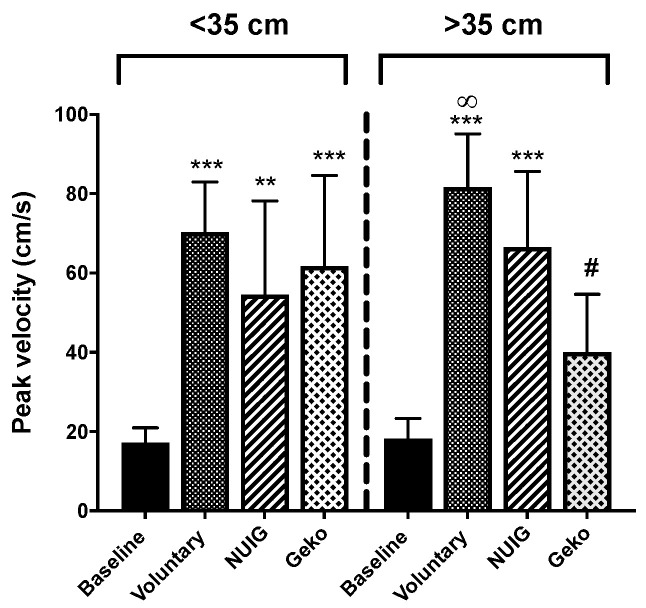
Effect of calf muscle circumference size on peak velocity. Voluntarily contraction (*n* = 10), Geko (*n* = 3–5), and NUIG (*n* = 4–5) showed a significant increase on PV in comparison to baseline (*n* = 10) for calf muscle circumference of <35 cm. For calf muscle of > 35 cm, only voluntarily contraction and NUIG showed a significant change in PV when compared to baseline. (***P* < 0.001, ****P* < 0.0001 versus baseline, #*P* < 0.05 versus. NUIG, ∞*P* < 0.001 versus Geko). Data shown as ± SD.

**Figure 6 jpm-10-00036-f006:**
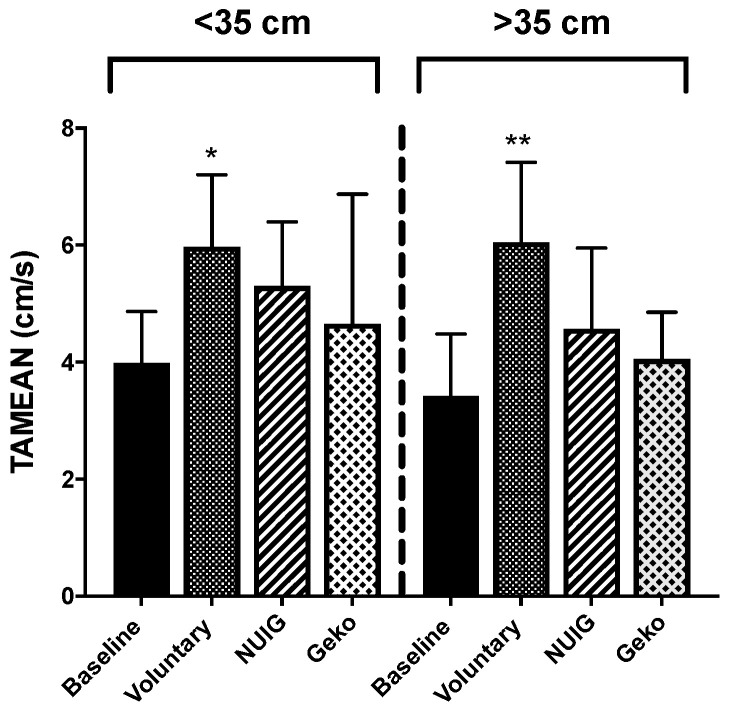
Effect of calf muscle circumference size on TAMEAN. For both calf muscle circumference of < 35 and >35 cm, voluntarily contraction (n = 10) only showed a significant increase on TAMEAN in comparison to baseline (*n* = 10). NUIG (*n* = 5) and the Geko device (*n* = 3–4) showed no alteration. (**P* < 0.05, ***P* < 0.001 versus baseline). Data shown as ± SD.

**Figure 7 jpm-10-00036-f007:**
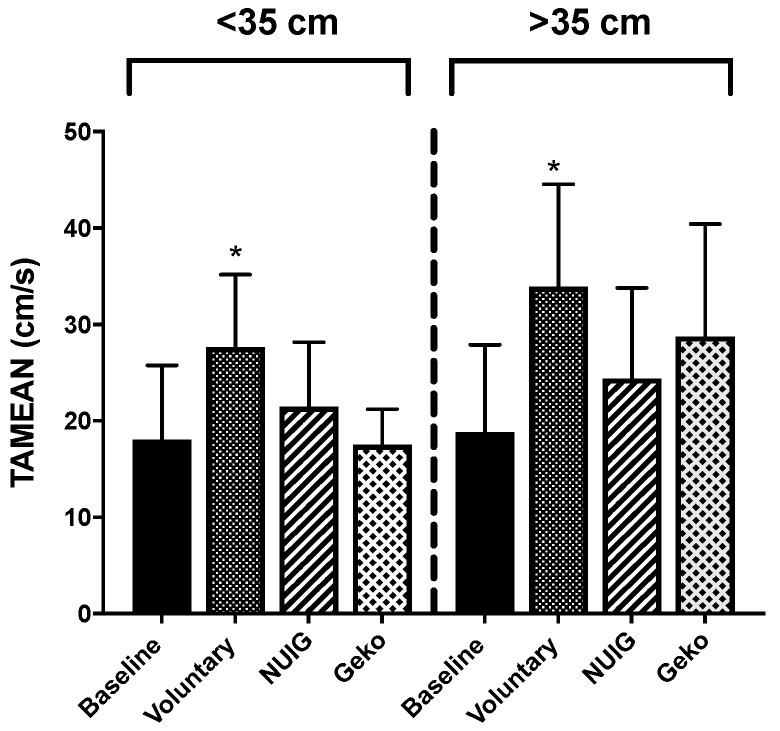
Effect of calf muscle circumference on the ejected volume. For a calf muscle circumference of both < 35 and >35 cm, only voluntarily contraction (*n* = 10) showed a significant increase in comparison to baseline (*n* = 10). There was no alteration in NUIG (*n* = 5) and Geko device (*n* = 3–4). (**P* < 0.05 versus baseline). Data shown as ± SD.

**Table 1 jpm-10-00036-t001:** Summary of results from all interventions and parameters measured.

Intervention	EV (mL)	P Versus. Baseline	PV (cm/s)	P Versus. Baseline	TAMEAN (cm/s)	P Versus. Baseline
**Baseline**	16.62 ± 9.1		17.54 ± 4.1		3.57 ± 1.0	
**Voluntary** **Contraction**	30.65 ± 13.7	*P* = 0.0004	75.91 ± 13.8	*P* < 0.0001	7.17 ± 3.0	*P* < 0.0001
**NUIG**	23.14 ± 11.4	*P* = 0.38	63.82 ± 18.2	*P* < 0.0001	5.34 ± 1.6	*P* = 0.08
**Geko^TM^**	19.76 ± 10.6	*P* = 0.90	46.34 ± 18.3	*P* < 0.0001	4.2 ± 1.6	*P* = 0.86

Ejected volume (EV), peak venous velocity (PV), and timed average velocity (TAMEAN) represented in subgroups for 12-s measurement windows. *P*-value versus baseline for intervention subgroup. *P*-value versus mean ± standard deviation.

**Table 2 jpm-10-00036-t002:** Summary of results from all interventions and parameters measured.

Calf Circumference	EV (mL)	P Versus. <35 cm	PV (cm/s)	P Versus. <35 cm	TAMEAN (cm/s)	P Versus. <35 cm
**<35 cm** **Baseline**	18.05 ± 7.7		17.16 ± 3.8		3.99 ± 0.9	
**<35 cm** **Voluntary** **Contraction**	27.63 ± 7.5	*P* = 0.0280	70.28 ± 12.7	*P* < 0.0001	5.98 ± 1.2	*P* < 0.0088
**<35 cm** **NUIG**	21.46 ± 6.7	*P* = 0.8166	54.55 ± 23.7	*P* = 0.0012	5.30 ± 1.01	*P* = 0.2560
**<35 cm** **Geko^TM^**	17.55 ± 3.7	*P* = 0.9994	61.72 ± 22.9	*P* < 0.0001	4.66 ± 2.2	*P* = 0.8095
**>35 cm** **Baseline**	18.8 ± 9.1		18.21 ± 5.1		3.42 ± 1.1	
**>35 cm** **Voluntary** **Contraction**	33.93 ± 10.6	*P* = 0.0119	81.60 ± 13.5	*P* < 0.0001	6.05 ± 1.4	*P* = 0.0004
**>35 cm** **NUIG**	24.41 ± 9.4	*P* = 0.7355	66.45 ± 19.2	*P* < 0.0001	4.57 ± 1.4	*P* = 0.3388
**>35 cm** **Geko^TM^**	28.74 ± 11.7	*P* = 0.4444	40.04 ± 14.6	*P* = 0.0627	4.06 ± 1.8	*P* = 0.8590

Ejected volume (EV), peak venous velocity (PV), and timed average velocity (TAMEAN) represented in subgroups for 12-s measurement windows. *P*-value versus baseline for intervention subgroup. P-value versus mean ± standard deviation.

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
