# Peer review of "Comparison of the Hemodynamic Performance of Two Neuromuscular Electrical Stimulation Devices Applied to the Lower Limb"

_jpm, 2020, doi:10.3390/jpm10020036_

Round 1
Reviewer 1 Report
Very important question
two thinks need clarification
- Main stay for venous ulcers is interventional treatment to incompetent veins rather than compression {EVRA and ESCHAR trials}
- in the experiments the timing of measurements from commencement to time of measurement need clarification. This is key as there may be some plateauing in flow after maximal stimulation over a certain point.
Author Response
We would like to thank the reviewer for the constructive feedback on our work. We will address each of the comments on a point by point basis below.
1. Main stay for venous ulcers is interventional treatment to incompetent veins rather than compression {EVRA and ESCHAR trials}
Response
We agree with the reviews that the landscape is changing as regards venous ulcer therapy, however, the guidelines are evolving. While the although the ESCHAR trial published in 2009, showed surgical ablation of superficial veins reduced ulcer recurrence, surgery did not improve healing. The European Society for Vascular Surgery (ESVS) most recent guidelines (2015) still recommend compression as a key element of venous disease management. Indeed, we agree the more recent EVRA now provides clear evidence that ablation accelerates healing and reduces recurrence. However, there is a note of caution here, of the 6500 patients assessed for inclusion, only 450 were included in the EVRA trial. So, while EVRA is an important, trial, longer-term follow-up data is required before definitive guidance can be revised. Thus, right now compression is still a cornerstone of venous therapy and this paper explores a potential adjunct that may enhance ulcer healing. It could be reasonable to assume that even with these new approaches, graduated compression bandaging will still be used in parallel and probably will continue to be over the period that the ulcer is.
All that said we were remiss not to reference the EVRA and ESCHAR trials and have now amended the manuscript to reflect this lines 48-54.
2. in the experiments the timing of measurements from commencement to time of measurement need clarification. This is key as there may be some plateauing in flow after maximal stimulation over a certain point.
Response
We thanks the reviewer for identifying that omission we have included the following line 134-136 “The measurement window for all test conditions, included 1 second before stimulation, and 11 seconds of blood flow post-stimulation onset”
Reviewer 2 Report
Manuscript Title: Comparison of the hemodynamic performance of two 2 neuromuscular electrical stimulation devices applied to the lower limb
Manuscript Number: jpm-779265
I congratulate the authors for a very interesting, well designed study and written manuscript. I have the following questions for the authors.
*1. Other neuromuscular devices are available in the market. Are these two devices tested unique and specifically designed for patient with chronic venous disease and specifically venous leg ulcers?
*2. Under Methods NUI Galway Device Protocol the authors go directly to Figure 3, before Figure 1 and 2, this is out of sequence, please correct.
*3. Under Methods in Hemodynamic Measurements, the authors should place reference(s) when indicating the benefits of maximum (peak) venous velocity information, and the benefits of increased venous ejection volume to support their statements (published literature on both topics in PubMed).
*4. In Figure 1, it would be important to label the peak venous velocity and TAMEAN velocities.
5. Of the parameters measured only PV was significantly increased for the neuromuscular devices compared to baseline. Of the three parameters measured which would be the most sensitive and important to demonstrate clinical effectiveness? Because the EV is not significantly different with electrical stimulation compared to baseline, would this imply that the electrical stimulations devices are likely not to achieve significant hemodynamic changes in patients with chronic venous disorders? In addition, the NMES do not replicate completely and significantly (compared to baseline) the physiologic properties that natural calf compression performs. Why is this? Are these devices NMES truly an important adjuvant for patients with venous leg ulcers, as much of the Introduction aims to solve?
*6. Under the Results heading 3.2, is misleading. For Figure 3 and Table 1, the TAMEAN for the NMES is not significant. In Table 1 there is no difference, as well as in the legend of Figure 3. Please address and clarify for consistency with the data.
7. How was the dominant and non-dominant limb determined in normal volunteers?
*8. Much of what NMES claims to do, an elastic compression stocking does the same and likely at a significant cost reduction. What then is the benefit of NMES? In addition, if a patient is not able to apply a elastic compression stocking of 40-50 mm Hg, there are several donning and doffing devices, and also Velcro strapped inelastic compression devices that are very effective.
*9. Do NMES actually activate calf muscle pump system? If this was entirely true would you not expect all parameters measured to be significantly improved as natural calf compression? I believe that the authors should use caution in their statements, since these devices have not been uniformly tested in patients with venous leg ulcers and demonstrating a clear and significant benefit.
10. Can NMES be used simultaneously with elastic compression stockings? Is there a synergistic benefit and any data for clinical improvement?
*11. Were there any side effects or complaints from the participants in this study due to the electrical stimulation? This information should be included.
12. Why would the authors test a NMES that stimulates muscle contraction in the anterior and lateral compartments, when most of the calf muscle pump function is driven by the soleus and gastrocnemius muscle groups?
*13. The authors should clearly list the limitations of this study.
Author Response
We would like to thank the reviewer for the constructive feedback on our work. We will address each of the comments on a point by point basis below.
*1. Other neuromuscular devices are available in the market. Are these two devices tested unique and specifically designed for the patient with chronic venous disease and specifically venous leg ulcers?
Response: The GEKO device is a commercially available stimulator and marketed for VTE prevention. It has guidance from NICE “The case for adopting the geko device is supported for use in people who have a high risk of venous thromboembolism and for whom other mechanical and pharmacological methods of prophylaxis are impractical or contraindicated. ” In addition, GEKO has also been used in a number of venous leg ulcer healing trials.
The NUIG is a research stimulator not commercially available but has been used in a number of small trials for venous ulcer healing and VTE prevention. Thus we are comparing our research grade device with the only readily commercially available NICE approved muscle stimulator.
*2. Under Methods NUI Galway Device Protocol the authors go directly to Figure 3, before Figure 1 and 2, this is out of sequence, please correct.
Response: Apologies for this typo, we have now corrected this in the text.
*3. Under Methods in Hemodynamic Measurements, the authors should place reference(s) when indicating the benefits of maximum (peak) venous velocity information, and the benefits of increased venous ejection volume to support their statements (published literature on both topics in PubMed).
Response: We have now included references for these statements
*4. In Figure 1, it would be important to label the peak venous velocity and TAMEAN velocities.
Response: We have now updated the legend of Figure 1.
5. Of the parameters measured only PV was significantly increased for the neuromuscular devices compared to baseline. Of the three parameters measured which would be the most sensitive and important to demonstrate clinical effectiveness? Because the EV is not significantly different with electrical stimulation compared to baseline, would this imply that the electrical stimulations devices are likely not to achieve significant hemodynamic changes in patients with chronic venous disorders? In addition, the NMES do not replicate completely and significantly (compared to baseline) the physiologic properties that natural calf compression performs. Why is this? Are these devices NMES truly an important adjuvant for patients with venous leg ulcers, as much of the Introduction aims to solve?
Response: This is a really great and thoughtful question(s). The answer is that it is too difficult to say at this point if anyone parameter is more clinically relevant better than any other. What is fair to assume is that given a voluntary contraction is the “best” solution to off-setting venous hypertension, the next best is a solution that which approaches a voluntary contraction. In this study, the NUIG device produces a blood flow waveform shape similar to that of a voluntary contraction. In the case of EV there are a number of factors to consider. In previous work, we used a 1 second measurement widow (Broderick et al., 2014, here only the active pulse was assessed) in this case the EV for NUIG NMES was significantly greater than baseline. In the current study, we choose a measurement window of 12 seconds to better reflect a real life use case where the user would wear the device for up to 30-60 minutes per day. In this case, there would be a trade-off between increased blood flow and developing muscle fatigue. The current study shows that a single large pulse (NUIG) or 12 smaller pulse (GEKO) is not sufficient. The including a two or three large pulse per 12 second window could improve blood flow but the effect of muscle fatigue would need to be investigated.
Clearly the voluntary contraction produced the greatest increased in blood flow due most likely to the well-coordinate, and appropriate activation of muscle groups in the lower limb. NMES is currently not able to achieve the level of coordination and selection specificity of a voluntary contraction. In addition, we normalised the position of the electrodes using surface anatomical landmarks. However, there is likely to be variability in neural pathways and depth thus introducing variability contraction efficiency in response to stimulation.
Are these devices truly an important adjuvant for patients? We would say that based on current evidence a tentative yes, but clearly there is a significant work to do to optimise the time of stimulation frequency, amplitude and potentially electrode placement. However even based on current settings if we consider a use case where the user has chronic venous disease and leads a very sedentary lifestyle the addition of an NMES therapy will provide benefit in terms of blood flow and improved muscle tone.
*6. Under the Results heading 3.2, is misleading. For Figure 3 and Table 1, the TAMEAN for the NMES is not significant. In Table 1 there is no difference, as well as in the legend of Figure 3. Please address and clarify for consistency with the data.
Response: This is an error and has been correct in the text.
7. How was the dominant and non-dominant limb determined in normal volunteers?
Response: The participants were skimpily asked to indicted which was there dominant limb.
*8. Much of what NMES claims to do, an elastic compression stocking does the same and likely at a significant cost reduction. What then is the benefit of NMES? In addition, if a patient is not able to apply a elastic compression stocking of 40-50 mm Hg, there are several donning and doffing devices, and also Velcro strapped inelastic compression devices that are very effective.
Response: This is a really good question and important to address. NMES produces a pulsatile physiological contraction of the calf muscles which is not achieved with graduated compression bandaging. In addition a big issue with the use of compression stockings is their use is affected by improper usage/application and low compliance rates. Patients commonly complain about compression stockings, stating that they are “cutting off circulation” and are “too hot”, “too itchy” and uncomfortable to wear (Donnelly T, McNeely B. The shocking stocking audit: an audit on the use of thromboembolic deterrent stockings (TEDS) for patients having surgery at Sligo regional hospital. J Perioper Pract. 2015;25:83-86. Anglen JO, Goss K, Edwards J, Huckfeldt RE. Foot pump prophylaxis for deep venous thrombosis: the rate of effective usage in trauma patients. Am J Orthop (Belle Mead NJ). 1998;27:580-582.). As for NMES, albeit in the small number of studies performed the compliance rates with the use of NMES have been favourable, with a high compliance rates reported as high as 100%±30% (some people used the device for longer than necessary) over a period of 7 days. Compliance rates with a prophylaxis device are dependent on several factors: patient-perceived comfort, satisfaction, usability and importance of the prophylaxis method in use and also whether or not the device has automatic built-in compliance monitoring and whether or not the patient is aware that their compliance is being recorded.
*9. Do NMES actually activate calf muscle pump system? If this was entirely true would you not expect all parameters measured to be significantly improved as natural calf compression? I believe that the authors should use caution in their statements, since these devices have not been uniformly tested in patients with venous leg ulcers and demonstrating a clear and significant benefit.
Response: We and others who work in the field would say that yes NMES does activate the calf muscle pump system. Of course as mentioned under point 5 above the effectiveness of NMES in terms of matching a voluntary contraction is complicated by issues of skin resistance, anatomy, neural pathways etc. We agree that these device have not been sufficiently tested but with studies such as ours and with optimisation the potential for benefits to patients exists.
10.Can NMES be used simultaneously with elastic compression stockings? Is there a synergistic benefit and any data for clinical improvement?
Response: We are currently running a study (total knee arthroplasty) for DVT prevention using an NMES device where the NMES electrodes are place under a conventional GCS where it appears to be working very well.
*11. Were there any side effects or complaints from the participants in this study due to the electrical stimulation? This information should be included.
Response: No side effects were reported by participants.
12. Why would the authors test a NMES that stimulates muscle contraction in the anterior and lateral compartments, when most of the calf muscle pump function is driven by the soleus and gastrocnemius muscle groups?
Response: Several authors have used various stimulation sites to enhance lower limb blood outflow including sites over the motor points of the gastrocnemius, soleus, combined gastrocnemius and soleus and the tibialis anterior in additional to peroneal nerve stimulation. There is no consistency or gold standard adopted as the chosen stimulation site.
The GEKO device is designed to be placed over the common peroneal nerve, at the back of the knee as this nerve is related to the contraction of muscles of the venous muscle pump of the lower Limb
The NUIG device uses two electrodes placed over the soleus muscle, this placement is based on previous work by Breen et al., 2012 and 2015. Authors systematically investigated all possible stimulation sites to find an optimal single- or two-site stimulation setup based on blood flow enhancement and fine wire EMG responses to stimulation.
*13. The authors should clearly list the limitations of this study.
Response: We have now included a limitations section